# OpenReview forum: "MM-BrowseComp: A Comprehensive Benchmark for Multimodal Browsing Agents"
_ICLR.cc/2026/Conference — ICLR 2026 Conference Withdrawn Submission_

### Official Review · Reviewer_dmLk · 2025-10-25

**Soundness:** 2
**Presentation:** 3
**Contribution:** 2
**Rating:** 4
**Confidence:** 4

**Summary:**

MM-BrowseComp consists of 224 challenging, hand-crafted questions distributed across 22 distinct subtasks covering five broad categories (Media, Technology, Society, Geography, and Academics). The questions are intentionally multi-hop and difficult, with a construction criteria ensuring they remain unanswerable by strong VLMs with web search in a single attempt, or by unfamiliar human annotators within five minutes. A crucial component is the verified checklist provided for each question, which represents the minimal irreducible reasoning path required to reach the correct answer. This checklist enables the use of Strict Accuracy (SA) alongside Overall Accuracy (OA), allowing for a fine-grained analysis that distinguishes genuine reasoning from "lucky guessing". Experimental results demonstrate that the top performer, OpenAI o3 with tools, achieved only 29.02% OA and 19.64% SA, confirming the benchmark’s challenging nature.

**Strengths:**

- The benchmark successfully bridges the gap left by previous textual benchmarks (like the original BrowseComp).
- The checklists provide fine-grained evaluation, moving beyond simple correctness to assess the path taken.
- Evaluates 18 models across multiple dimensions with detailed error taxonomy and modality-specific performance breakdown.

**Weaknesses:**

- While the authors convincingly justify the size through the rigor of construction and high filtering rate, a total of 224 instances across 22 distinct subtasks may be insufficient for reporting reasonable scores at this granularity.
- Heavy reliance on GPT-4o-2024-11-20 as the sole evaluator for checklist, and I believe this might add certain evaluation bias.

**Questions:**

- Could the authors elaborate on the strict criteria used by annotators to ensure the reasoning checklist is truly "irreducible"?
- Why evaluate open-source agents on only 54 instances? This seems too limited for reliable conclusions. What were the selection criteria?

---

> ### Author Response · Authors · 2025-11-26
> **Official Comment by Authors (1/3)**
>
> We sincerely appreciate your valuable comments. We are encouraged by your recognition that **MM-BrowseComp addresses the lack of multimodal browsing benchmarks while offering fine-grained checklist evaluations**. We will address your concerns in detail below.
>
> > W1: While the authors convincingly justify the size through the rigor of construction and high filtering rate, a total of 224 instances across 22 distinct subtasks may be insufficient for reporting reasonable scores at this granularity.
>
> A: Thank you for raising this issue about dataset granularity. We would like to clarify how we intend MM-BrowseComp to be used and why we believe 224 instances are appropriate for our goals:
>
> 1. **Our main conclusions are drawn at the dataset level, not the subtask level**. In the current version, all key results, such as Overall Accuracy (OA), Strict Accuracy (SA), and Average Checklist Score (AVG CS), are reported over the  benchmark rather than for each subtask individually. Subtasks are primarily used as analytical lenses, rather than as 22 independent sub-benchmarks with separate leaderboards.
>
> 2. **Subtasks are designed to ensure balanced coverage and diversity across the dataset**. The 22 subtasks are not intended to serve as 22 independent sub-benchmarks, but rather as a taxonomy that ensures the dataset construction process covers wide and different multimodal browsing and reasoning scenarios (e.g., media, technology, society, geography, and academic content) and facilitating the recruitment of human experts with corresponding domain knowledge. The primary role of this taxonomy is to ensure that models are exposed to a wide range of task formats and domains, instead of being evaluated on a narrow slice of multimodal search scenes.
>
> 3. **The overall size of MM-BrowseComp is comparable to widely used agent benchmarks**. For example, GAIA prioritizes carefully constructed problems over large-scale synthetic data, comprising 466 total instances with a validation set of just 165 questions (53 in Level 1, 86 in Level 2, and 26 in Level 3).  Likewise, BEARCUBS includes only 111 tasks, yet it is still considered a valuable benchmark for assessing GUI / computer-use agents. MM-BrowseComp follows the same design philosophy in the context of multimodal browsing agent: the number of instances is not extremely large, but each sample is carefully constructed and filtered.  Moreover, given the high computational and time cost of evaluating browsing agents on such tasks, this design strikes a practical balance between construction quality and evaluation cost.
>
> We provide additional examples in the table below.
>
> | Benchmark          | Venue         | # Instances       | # Categories | Reference |
> |--------------------|---------------|-------------------|-------------|-----------|
> | GAIA               | ICLR 2024     | 165 (val set)     | 3           | https://iclr.cc/virtual/2024/poster/18176 |
> | MMSearch           | ICLR 2025     | 300               | 14          | https://iclr.cc/virtual/2025/poster/30134 |
> | AgentHarm          | ICLR 2025     | 440               | 11          | https://iclr.cc/virtual/2025/poster/32106 |
> | OSWorld            | NeurIPS 2024  | 369               | 4           | https://neurips.cc/virtual/2024/poster/97468 |
> | Mind2Web 2         | NeurIPS 2025  | 130               | -           | https://neurips.cc/virtual/2025/loc/san-diego/poster/121798 |
> | Open CaptchaWorld  | NeurIPS 2025  | 225               | 20          | https://neurips.cc/virtual/2025/poster/121537 |
> | BEARCUBS           | COLM 2025     | 111               | 1           | https://openreview.net/forum?id=0JzWiigkUy |

---

> ### Author Response · Authors · 2025-11-26
> **Official Comment by Authors (2/3)**
>
> > W2: Heavy reliance on GPT-4o-2024-11-20 as the sole evaluator for checklist, and I believe this might add certain evaluation bias.
>
> A: Thank you for raising this concern. We would like to  clarify why we believe our use of an LLM-as-a-judge is acceptable, and how we have assessed its robustness.
>
> First, in our benchmark, we only use the LLM-as-a-judge **for objective tasks with clear ground truth**:
> - deciding whether a model’s final answer is correct by comparing it with the ground truth answer,
> - deciding, for each checklist item, whether the agent’s trajectory satisfies the corresponding requirement.
>
> These decisions are made with respect to a ground truth answer and an explicit checklist, rather than open-ended or subjective criteria (e.g., helpfulness). In particular, our ground truth answers are short, verifiable phrases (e.g., entity names, dates, or numbers) rather than long free-form responses, which **makes the judging task much easier and less ambiguous**. For a strong model like GPT-4o-2024-11-20, we believe such constrained judgments are straightforward and reliable.
>
> Second, our evaluation methodology aligns with recent popular benchmarks, such as BrowseComp[1] and Humanity's Last Exam (HLE)[2], which employ LLMs as automated judges. Following this paradigm, we use a single strong model to perform semantically grounded judgments over short answers and checklist items, which we believe is **in line with current community practice**.
>
> To further assess potential evaluation bias by relying on GPT-4o-2024-11-20 as judge, we **re-evaluated all tool-free VLMs and tool-augmented VLMs by using Gemini-2.5-Flash** as an alternative LLM judge. Gemini-2.5-Flash is given the same evaluation prompts. The resulting scores are relatively consistent with those obtained from GPT-4o-2024-11-20. While there are small numerical differences, the overall trends and our main conclusions remain unchanged.
>
> **Tool-free VLMs (GPT-4o-2024-11-20 vs. Gemini-2.5-Flash as judges)**
>
> | Model                              | OA (%) by GPT-4o-2024-11-20 | SA (%) by GPT-4o-2024-11-20 | AVG CS (%) by GPT-4o-2024-11-20 | OA (%) by Gemini-2.5-Flash | SA (%) by Gemini-2.5-Flash | AVG CS (%) by Gemini-2.5-Flash |
>    | ---------------------------------- | ------------- | ------------- | ----------------- | ------------- | ------------- | ----------------- |
>    | o4-mini-high                       | 7.14          | 3.13          | 13.67             | 8.04          | 2.68          | 14.33             |
>    | o4-mini                            | 5.36          | 2.23          | 12.41             | 6.25          | 1.34          | 15.17             |
>    | GPT-4.1                            | 7.59          | 5.36          | 14.68             | 8.48          | 3.57          | 18.45             |
>    | GPT-4o-2024-11-20                  | 1.34          | 0.45          | 4.63              | 1.79          | 0.45          | 5.53              |
>    | GPT-4o-mini                        | 0.89          | 0.00          | 1.47              | 1.34          | 0.00          | 1.69              |
>    | Gemini-2.5-Pro-Preview-05-06       | 6.31          | 4.50          | 11.56             | 11.16         | 6.25          | 21.01             |
>    | Gemini-2.5-Flash-Preview-05-20     | 2.70          | 2.25          | 8.57              | 4.02          | 4.02          | 16.22             |
>    | Qwen2.5-VL-72B-Instruct            | 0.45          | 0.00          | 3.58              | 0.89          | 0.00          | 4.64              |
>    | Qwen2.5-VL-32B-Instruct            | 1.45          | 0.00          | 1.77              | 1.79          | 0.00          | 4.01              |
>    | Qwen2.5-VL-7B-Instruct             | 0.00          | 0.00          | 0.15              | 0.00          | 0.00          | 0.89              |
>    | Llama-4-Maverick-17B-128E-Instruct | 2.68          | 0.45          | 6.09              | 4.91          | 0.45          | 6.61              |
>
> **Tool-augmented VLMs (GPT-4o-2024-11-20 vs. Gemini-2.5-Flash as judges)**
>
>  | Model                          | OA (%) by GPT-4o-2024-11-20 | SA (%) by GPT-4o-2024-11-20 | AVG CS (%) by GPT-4o-2024-11-20 | OA (%) by Gemini-2.5-Flash | SA (%) by Gemini-2.5-Flash | AVG CS (%) by Gemini-2.5-Flash |
>    | ------------------------------ | ------------- | ------------- | ----------------- | ------------- | ------------- | ----------------- |
>    | o3                             | 29.02         | 19.64         | 36.49             | 29.02         | 19.64         | 40.90             |
>    | Gemini-2.5-Pro-Preview-05-06   | 7.14          | 3.57          | 15.21             | 8.93          | 4.91          | 22.58             |
>    | Gemini-2.5-Flash-Preview-05-20 | 3.12          | 3.12          | 11.34             | 4.02          | 3.12          | 15.71             |
>
>
> [1] Browsecomp: A simple yet challenging benchmark for browsing agents, OpenAI, arxiv, 2025.
>
> [2] Humanity's last exam. Scale AI. arXiv. 2025.

---

> ### Author Response · Authors · 2025-11-26
> **Official Comment by Authors (3/3)**
>
> > Q1: Could the authors elaborate on the strict criteria used by annotators to ensure the reasoning checklist is truly "irreducible"?
>
> A: Thank you for your question. In practice, we ensure this property using the following criteria:
>
> First, after constructing a question and its checklist, annotators iteratively prune redundant items: for each item, whether an ideal browsing agent could still reliably obtain the correct answer if that item were removed; if so, the item is deleted, and this process is repeated until no remaining step can be removed.
>
> Second, every checklist item must be tied to explicit evidence from the browsing process (a specific web page, image, or video) and annotated with its primary modality (text, image, or video). In the final benchmark, we obtain 658 checklist items, with 36.3% text-based, 35.0% image-based, and 28.7% video-based steps, ensuring that each item is grounded in concrete fact rather than being hypothetical.
>
> Finally, as described in Section 3.2, our three-stage verification pipeline includes multiple rounds of human review over the question, ground truth answer, and checklist; during these rounds, annotators again remove any step that is unnecessary.
>
> Overall, these measures are designed to make the checklist as irreducible as possible in practice.
>
> ---
>
> > Q2: Why evaluate open-source agents on only 54 instances? This seems too limited for reliable conclusions. What were the selection criteria?
>
> A: Thank you for your nice question. Regarding this concern, we clarify our design choices as follows:
>
> Running an open-source agent framework on MM-BrowseComp is more expensive than a single model inference: each question typically involves hundreds of internet search requests, several rounds of tool invocations, and long-context reasoning loops on our own server. Consequently, evaluating every open-source agent on all 224 instances is very computationally and time-intensive, and a single end-to-end evaluation of the full set requires approximately 6-8 hours.
>
> As a compromise, we adopt a balanced sampling strategy: we perform uniform sampling over the 22 subtasks, randomly selecting a small number of instances from each subtask and obtaining a set of 54 instances that still covers all subtask types. All open-source agents are evaluated on the same 54-instance subset, ensuring a fair comparison. This scale is comparable to the subset sizes commonly used in other agent benchmarks: for example, the GAIA validation set contains only 53 Level-1 and 26 Level-3 questions.
>
> To verify the consistency between the subset and the full dataset, **during the rebuttal phase, we additionally evaluated open-source agents on the full 224-instance benchmark**. The results are shown below and lead to the same qualitative picture: all open-source agents achieve low OA/SA/AVG CS scores, clearly below o3, and the relative differences among them **are consistent with the subset evaluation**.
>
> **Open-source agents on the full MM-BrowseComp benchmark (224 instances)**
>
> | Agent      | Model                           | OA (%) | SA (%) | AVG CS (%) |
> |-----------|----------------------------------|--------|--------|------------|
> | Agent-R1  | Gemini-2.5-Flash-Preview-05-20  | 6.25   | 4.46   | 8.13       |
> | Agent-R1  | GPT-4o-2024-11-20               | 4.46   | 4.46   | 7.32       |
> | Agent-R1  | Qwen2.5-VL-72B-Instruct         | 2.68   | 0.45   | 3.99       |
> | OWL       | Gemini-2.5-Flash-Preview-05-20  | 6.25   | 0.89   | 4.47       |
> | OWL       | GPT-4o-2024-11-20               | 3.13   | 0.89   | 11.65      |
> | OWL       | Qwen2.5-VL-72B-Instruct         | 2.68   | 0.45   | 2.49       |
> | DeerFlow  | Gemini-2.5-Flash-Preview-05-20  | 2.68   | 2.68   | 3.58       |
> | DeerFlow  | GPT-4o-2024-11-20               | 3.13   | 2.68   | 5.30       |
> | DeerFlow  | Qwen2.5-VL-72B-Instruct         | 2.68   | 0.89   | 5.97       |
> | WebDancer | Gemini-2.5-Flash-Preview-05-20  | 2.68   | 2.68   | 4.20       |
> | WebDancer | GPT-4o-2024-11-20               | 3.57   | 2.68   | 3.83       |
> | WebDancer | Qwen2.5-VL-72B-Instruct         | 0.89   | 0.45   | 1.30       |
> | WebDancer | WebDancer-32B                   | 2.68   | 0.89   | 2.80       |
>
> Overall, the **54-instance subset is a cost-effective way** to obtain **qualitative conclusions** about open-source agents, and the full-benchmark results confirm that these conclusions continue to hold when evaluating on all 224 instances.

---

### Official Review · Reviewer_WJHa · 2025-11-01

**Soundness:** 2
**Presentation:** 3
**Contribution:** 2
**Rating:** 2
**Confidence:** 3

**Summary:**

The paper introduces MM-BrowseComp for evaluating agents that must browse the web and reason over multimodal content. Each instance comes with an irreducible reasoning checklist that specifies the minimal sequence of retrieval and reasoning steps required to reach the answer, enabling fine-grained assessment beyond final-answer accuracy. On this benchmark, strong systems achieve only ~29% accuracy, underscoring the challenge and current gap in native multimodal browsing.

**Strengths:**

1. Clarity: The paper is readable and well-structured, with intuitive examples and comprehensive task taxonomy/mixture. Construction principles and validation steps are communicated with sufficient detail.

2. Significance: Addresses a timely need: deep web browsing with native multimodality—central for real-world assistants. The results and analyses (e.g., modality-specific performance, test-time scaling, error taxonomy) are likely to shape evaluation practices and agent design.

**Weaknesses:**

1. Scale: 224 instances is on the small side for a general-purpose benchmark spanning 22 subtasks; per-subtask sample sizes are too thin for robust statistics. Consider releasing a larger dev/test split or staged expansions, and report confidence intervals (e.g., bootstrap over items) in the main text. The dataset probably won't be very meaningful if the data size is too small.

2. Potential construction bias and leakage checks. During dataset construction, there could be several stages with risk of potential biases. Difficulty criteria include “unanswerable by strong models in one attempt.” can be subjective. The dataset construction lacks inter-annotator agreement. This risks encoding model-specific blind spots. Add contamination audits, time-stamped sources, and a multi-attempt human check protocol report (agreement, time-to-solve).

3. While the paper has some analysis on the evaluation results, the LLM-judge based analysis seems not very scalable if the data size grows large. Also the llm judge backbone may also introduce extra bias for analysis.

**Questions:**

1. Checklist design & validation. How do you ensure minimality and non-redundancy of checklists across annotators? Report inter-annotator agreement on checklists and provide a public rubric of what counts as “completed.” (This would reduce reviewer subjectivity when others extend the benchmark.)

2. Tool standardization. Could you release a reference tool suite (OCR, layout/grounding, video frame sampler) and a tool-capability checklist per model/agent so results are not confounded by missing/different tools? This would also clarify where o3’s edge stems from (backbone vs. toolset).

3. Dataset growth & governance. Any plan for a continually updated MM-BrowseComp with frozen yearly snapshots and public leaderboards? Is there any plan to make the data collection pipeline more scalable and generalizable.

4. Error taxonomy reliability. The failure analysis uses GPT-4o to label errors. Please report labeling agreement (e.g., dual-judge consistency) and try to use different LLM to label the errors. Would GPT-4o have bias towards OpenAI models like o3?

---

> ### Author Response · Authors · 2025-11-26
> **Official Comment by Authors (1/5)**
>
> > W1: Scale: 224 instances is on the small side for a general-purpose benchmark spanning 22 subtasks; per-subtask sample sizes are too thin for robust statistics. Consider releasing a larger dev/test split or staged expansions, and report confidence intervals (e.g., bootstrap over items) in the main text. The dataset probably won't be very meaningful if the data size is too small.
>
> A: Thank you for your thorough and thoughtful feedback. In response to these concerns, we provide the following clarifications:
>
> Our primary focus is on the performance across the complete set of 224 instances, rather than fine-grained statistics on individual subtasks. All key conclusions in the paper, such as Overall Accuracy (OA), Strict Accuracy (SA), and Average Checklist Score (AVG CS), and performance differences between models, are reported and analyzed at the dataset level. **The 22 subtasks are primarily used to ensure diversity** in multimodal browsing and reasoning scenarios, not to serve as independent, statistically robust leaderboards for each subtask. We will further emphasize this point in the revised version of the paper.
>
> From a benchmark design perspective, the scale of MM-BrowseComp represents a strategic **trade-off between dataset size and computational cost**. Unlike standard single-model inference, executing agent frameworks is significantly more resource-intensive: each query typically involves hundreds of search requests, multiple rounds of tool invocation, and extensive long-context reasoning loops. Currently, a full evaluation of 224 questions requires 6–8 hours to complete. A larger dataset would essentially harm efficient iterative development.
>
> Consequently, **other prominent agent benchmarks also prioritize high-quality curation over data scale**. For instance, GAIA deliberately limits its scope to 466 total instances (with only 165 in the public validation set), and BEARCUBS remains a standard for GUI agents despite containing only 111 instances. MM-BrowseComp adheres to this established community standard, balancing rigorous evaluation with practical accessibility. We also include additional comparable benchmarks in the table below:
>
> | Benchmark          | Venue         | # Instances       | # Categories | Reference |
> |--------------------|---------------|-------------------|-------------|-----------|
> | GAIA               | ICLR 2024     | 165 (val set)     | 3           | https://iclr.cc/virtual/2024/poster/18176 |
> | MMSearch           | ICLR 2025     | 300               | 14          | https://iclr.cc/virtual/2025/poster/30134 |
> | AgentHarm          | ICLR 2025     | 440               | 11          | https://iclr.cc/virtual/2025/poster/32106 |
> | OSWorld            | NeurIPS 2024  | 369               | 4           | https://neurips.cc/virtual/2024/poster/97468 |
> | Mind2Web 2         | NeurIPS 2025  | 130               | -           | https://neurips.cc/virtual/2025/loc/san-diego/poster/121798 |
> | Open CaptchaWorld  | NeurIPS 2025  | 225               | 20          | https://neurips.cc/virtual/2025/poster/121537 |
> | BEARCUBS           | COLM 2025     | 111               | 1           | https://openreview.net/forum?id=0JzWiigkUy |

---

> ### Author Response · Authors · 2025-11-26
> **Official Comment by Authors (2/5)**
>
> > W2: Potential construction bias and leakage checks. During dataset construction, there could be several stages with risk of potential biases. Difficulty criteria include “unanswerable by strong models in one attempt.” This can be subjective. The dataset construction lacks inter-annotator agreement. This risks encoding model-specific blind spots. Add contamination audits, time-stamped sources, and a multi-attempt human check protocol report (agreement, time-to-solve).
>
> A: Thank you for your detailed suggestions. In response to your concerns, we provide the following detailed clarifications:
> 1. **Difficulty criteria and bias**. The difficulty criterion of “unanswerable by strong models in one attempt” is not a subjective judgment, but rather an **objective filtering criterion based on clear standards**. For each candidate question, we perform checks to ensure that both Gemini-2.5-Pro and GPT-4o, with access to web search tools, are unable to answer the question correctly in one attempt. Only then is the question retained.
> 2. **Inter-annotator consistency**. As described in Sec 3.2, we control for this through unified annotation guidelines, golden examples, and multiple rounds of review. Before the formal annotation process, all annotators are provided with the same instructions and examples, and we align granularity and style on a small batch of pilot and calibration tasks. Crucially, a dedicated core team conducted a comprehensive review of all questions, resolving any ambiguity through real-time discussion. Therefore, we **achieved nearly full internal consistency on the final dataset** through this rigorous validation process.
> 3. **Leakage checks and contamination audits**. To prevent data leakage, we follow the best practices from community. For instance, all key fields (question/answer/checklist) in our **dataset are encrypted** by default in the repository, and decryption occurs only during the evaluation phase. This prevents models from retrieving unencrypted raw data during the reasoning process. Additionally, we embed canary strings within the dataset to help filter out any data that might have been inadvertently exposed in future training corpora.
> 4. **Source reliability**. During data construction, annotators are instructed to prioritize **authoritative sources** and, when necessary, **include time constraints** on the questions to ensure the answers are not affected by temporal changes. This guarantees that answers remain consistent and reliable over time.
> 5. **Human baseline and time-to-solve**. To help evaluate model/agent performance, we conducted an evaluation as a human baseline. We recruited 6 AI master students as volunteers and allocated the entire MM-BrowseComp dataset to them in a shuffled manner. Each question had a time limit of 10 minutes, and they were allowed to use any tools except AI assistance. The results are as follows:
>
> | Metric               | OA(%)  | SA(%) | AVG CS(%) | Avg. Cost Time (mins) |
> |----------------------|--------|-------|--------|---------------------|
> | Human Performance    | 60.26 | 50.00   | 58.36 | 7.8                 |
>
>
> This evaluation indicates that **MM-BrowseComp tasks present challenges even for humans**. Furthermore, it demonstrates that there remains a **significant margin for improvement before agents can match human-level capabilities**.
>
> We hope these explanations help address the reviewer’s concerns about construction bias, leakage, and other related issues with the dataset.

---

> ### Author Response · Authors · 2025-11-26
> **Official Comment by Authors (3/5)**
>
> > W3: While the paper has some analysis on the evaluation results, the LLM-judge based analysis seems not very scalable if the data size grows large. Also the llm judge backbone may also introduce extra bias for analysis.
>
> A: Thank you for raising this concern regarding the scalability and potential bias of using an LLM-based judge. We would like to clarify why we believe this approach is feasible within the current dataset size and how we have considered bias in our analysis.
> 1. **Scalability issues**. At the current scale of MM-BrowseComp, which includes only 224 instances, we believe there are no scalability issues. The cost of using GPT-4o-2024-11-20 as the judge is relatively small compared to running a full browsing agent. Even if the dataset were to scale to thousands of instances, the cost of running the LLM-judge would still be manageable and remain secondary to the costs of running the agent itself.
> 2. **Bias problem**. In our benchmark, we only use the LLM-as-a-judge **for objective tasks with clear ground truth**:
> - deciding whether a model’s final answer is correct by comparing it with the ground truth answer,
> - deciding, for each checklist item, whether the agent’s trajectory satisfies the corresponding requirement.
> These judgements are made with respect to a ground truth answer and an explicit checklist, rather than open-ended or subjective criteria (e.g., helpfulness). In particular, our ground truth answers are short, verifiable phrases (e.g., entity names, dates, or numbers) rather than long free-form responses, which **makes the judging task much easier and less ambiguous**. For a strong model like GPT-4o-2024-11-20, we believe such constrained judgments are straightforward and reliable. Recent popular benchmarks, such as BrowseComp[1] and Humanity's Last Exam (HLE)[2] also employ LLMs as automated judges. Following this paradigm, we use a single strong model to perform semantically grounded judgments over short answers and checklist items, which we believe is **in line with current community practice**.
>
> To further assess potential evaluation bias by relying on GPT-4o-2024-11-20 as judge, we **re-evaluated all tool-free VLMs and tool-augmented VLMs by using Gemini-2.5-Flash** as an alternative LLM judge. Gemini-2.5-Flash is given the same evaluation prompts. The resulting scores are relatively consistent with those obtained from GPT-4o-2024-11-20: while there are small numerical differences, the overall trends and our main conclusions remain unchanged.
>
> **Tool-free VLMs (GPT-4o-2024-11-20 vs. Gemini-2.5-Flash as judges)**
> | Model                              | OA (%) by GPT-4o-2024-11-20 | SA (%) by GPT-4o-2024-11-20 | AVG CS (%) by GPT-4o-2024-11-20 | OA (%) by Gemini-2.5-Flash | SA (%) by Gemini-2.5-Flash | AVG CS (%) by Gemini-2.5-Flash |
>    | ---------------------------------- | ------------- | ------------- | ----------------- | ------------- | ------------- | ----------------- |
>    | o4-mini-high| 7.14| 3.13| 13.67| 8.04          | 2.68          | 14.33             |
>    | o4-mini| 5.36| 2.23| 12.41| 6.25| 1.34          | 15.17             |
>    | GPT-4.1| 7.59| 5.36| 14.68| 8.48          | 3.57          | 18.45             |
>    | GPT-4o-2024-11-20| 1.34| 0.45          | 4.63              | 1.79          | 0.45          | 5.53              |
>    | GPT-4o-mini| 0.89| 0.00| 1.47              | 1.34          | 0.00          | 1.69              |
>    | Gemini-2.5-Pro-Preview-05-06       | 6.31          | 4.50          | 11.56             | 11.16         | 6.25          | 21.01             |
>    | Gemini-2.5-Flash-Preview-05-20     | 2.70          | 2.25          | 8.57              | 4.02          | 4.02          | 16.22             |
>    | Qwen2.5-VL-72B-Instruct| 0.45| 0.00          | 3.58              | 0.89          | 0.00          | 4.64              |
>    | Qwen2.5-VL-32B-Instruct| 1.45| 0.00          | 1.77              | 1.79          | 0.00          | 4.01              |
>    | Qwen2.5-VL-7B-Instruct| 0.00| 0.00          | 0.15              | 0.00          | 0.00          | 0.89              |
>    | Llama-4-Maverick-17B-128E-Instruct | 2.68          | 0.45          | 6.09              | 4.91          | 0.45          | 6.61              |
>
> **Tool-augmented VLMs (GPT-4o-2024-11-20 vs. Gemini-2.5-Flash as judges)**
>
>  | Model| OA (%) by GPT-4o-2024-11-20 | SA (%) by GPT-4o-2024-11-20 | AVG CS (%) by GPT-4o-2024-11-20 | OA (%) by Gemini-2.5-Flash | SA (%) by Gemini-2.5-Flash | AVG CS (%) by Gemini-2.5-Flash |
>    | ------------------------------ | ------------- | ------------- | ----------------- | ------------- | ------------- | ----------------- |
>    | o3| 29.02| 19.64| 36.49| 29.02| 19.64| 40.90 |
>    | Gemini-2.5-Pro-Preview-05-06   | 7.14| 3.57| 15.21| 8.93| 4.91| 22.58|
>    | Gemini-2.5-Flash-Preview-05-20 | 3.12| 3.12| 11.34| 4.02| 3.12| 15.71|
>
> [1] Browsecomp: A simple yet challenging benchmark for browsing agents, OpenAI, arxiv, 2025.
>
> [2] Humanity's last exam. Scale AI. arXiv. 2025.

---

> ### Author Response · Authors · 2025-11-26
> **Official Comment by Authors (4/5)**
>
> > Q1: Checklist design & validation. How do you ensure minimality and non-redundancy of checklists across annotators? Report inter-annotator agreement on checklists and provide a public rubric of what counts as “completed.” (This would reduce reviewer subjectivity when others extend the benchmark.)
>
> A: Thank you for this suggestion. We take **three main approaches to reduce redundancy** of checklists.
>
> First, in our annotation guidelines, we require annotators to perform a step-by-step pruning check. After constructing a checklist, annotators are asked to evaluate whether removing any step would still allow an ideal agent, which relies solely on web evidence, to arrive at the correct answer. If the step can be removed or merged with another, it is discarded. This process is repeated until no further steps can be eliminated, ensuring that each checklist item is essential.
>
> Second, every checklist item must be directly linked to concrete evidence, such as a specific web page, image, or video, with the modality (text, image, or video) clearly marked. This ensures that each checklist item is grounded in verifiable content.
> Third, as described in Section 3.2, our multi-stage human review process involves repeated checks to remove redundant steps. This process explains why 25.3% of the 300 initial candidate questions were discarded, and 21.0% were significantly modified, with only 224 questions ultimately retained.
>
> Regarding inter-annotator consistency, we control for this through unified annotation guidelines, golden examples, and multiple rounds of review. Before the formal annotation process, all annotators are provided with the same instructions and examples, and we align granularity and style on a small batch of pilot and calibration tasks, as described in Sec 3.2. Crucially, a dedicated core team conducted a comprehensive review of all questions, resolving any ambiguity through real-time discussion. Therefore, we achieved nearly full internal agreement on the final dataset through this rigorous validation process.
>
> ---
>
>
> > Q2: Tool standardization. Could you release a reference tool suite (OCR, layout/grounding, video frame sampler) and a tool-capability checklist per model/agent so results are not confounded by missing/different tools? This would also clarify where o3’s edge stems from (backbone vs. toolset).
>
> A: Thank you for this suggestion. Our goal with MM-BrowseComp is to evaluate browsing agents in an end-to-end manner, rather than comparing the same backbone under a fully standardized toolset. In this setup, the specific tools that agents use (such as OCR, search APIs, etc.) are **intrinsic components of an agent framework which are decided by their developers** like OpenAI or Google. We believe that tool selection and usage, along with the agent's strategy, should be evaluated holistically, rather than being rigidly fixed across all methods.
>
> ---
>
>
> > Q3: Dataset growth & governance. Any plan for a continually updated MM-BrowseComp with frozen yearly snapshots and public leaderboards? Is there any plan to make the data collection pipeline more scalable and generalizable?
>
> A: Thank you for your valuable suggestions. Regarding the growth of the dataset, we plan to establish the current version as the first official snapshot (e.g., MM-BrowseComp-2025), with the goal of expanding the dataset in future releases through a staged approach. This will allow us to introduce new instances and tasks incrementally while ensuring high quality.
>
> In terms of the public leaderboard, we plan to implement a simple public leaderboard that will be accessible based on our open-source code and evaluation scripts. This will allow the broader community to report their results under a consistent protocol, fostering a reproducible comparison between models/agents. The leaderboard will be updated periodically with new evaluations of existing and future models.
>
> As for the scalability and generalizability of our data collection pipeline, we have already developed a general pipeline for constructing tasks in our paper, which can accommodate a wide range of new task types and domains. In future updates, we aim to encourage community contributions by allowing others to propose and contribute new task instances. These contributions will be reviewed and integrated into the dataset following our standard review process to ensure the tasks meet our quality standards. This will help us scale the dataset.
>
> By adopting this approach, we aim to ensure that the dataset grows in a manageable and sustainable way, while fostering community involvement and maintaining the high standards of quality required for reliable benchmarking.

---

> ### Author Response · Authors · 2025-11-26
> **Official Comment by Authors (5/5)**
>
> > Q4: Error taxonomy reliability. The failure analysis uses GPT-4o to label errors. Please report labeling agreement (e.g., dual-judge consistency) and try to use different LLM to label the errors. Would GPT-4o have bias towards OpenAI models like o3?
>
> A: Thank you for this good question. We want to emphasize that it is important to distinguish between our primary performance metric and failure analysis.
>
> First, our primary performance metrics, specifically Overall Accuracy (OA), Strict Accuracy (SA), and Average Checklist Score (AVG CS), are derived directly from evaluations against ground truth answers. Since these answers consist of short and verifiable phrases such as specific entities or dates, the evaluation process is highly objective and minimally susceptible to judge bias. We **validated our results by using GPT-4o and Gemini-2.5-Flash as dual-judge** (see W3). We observed **no significant divergence** in their judging results.
>
> Second, failure analysis is applied to incorrect agent response and GPT-4o is used only as a diagnostic tool. As such, any potential bias in error taxonomy does not affect the final conclusion of this paper.

---

### Official Review · Reviewer_3Njv · 2025-11-01

**Soundness:** 2
**Presentation:** 2
**Contribution:** 2
**Rating:** 4
**Confidence:** 4

**Summary:**

The authors introduce a new benchmark to evaluate web-browsing agents in multimodal environments, where text shortcuts are not available. It contains 224 carefully designed questions that often include images in the prompt or require agents to pull information from visuals on real web pages. Each question is paired with a verified checklist of reasoning steps and supporting evidence. The best system only achieves only about 29% accuracy. The findings show that current models still struggle significantly with multimodal browsing and reasoning.Models perform much worse on visual content than on text and tend to depend on shallow image-captioning shortcuts rather than genuine visual understanding.

**Strengths:**

- The data construction process ensures questions require multimodal browsing, effectively eliminating text shortcuts.
- Queries in the dataset go through rigorous difficulty-based filtering.
- The human-verified checklist of minimal finegrained reasoning steps provides a valuable signal, it provides a way for evaluation to go beyond just right/wrong final answers.

**Weaknesses:**

- There is missing a human baseline to calibrate what model accuracy means. It would provide an estimate for the performance ceiling of this task.
- In 3.1.1, authors assert that essential information to solve a task should not appear in any text source. However, there is no mention of how this verification is done.
- Although the authors repeatedly refer to “video-dependent” tasks, the paper never specifies how models are expected to engage with videos. Are agents intended to interact with video content directly, or are they simply expected to rely on accompanying textual information, like transcripts or descriptions?

**Questions:**

- Figure 3 could be clearer. Rather than focusing on whether each input includes an image, it might be more informative to present statistics on the actual modalities required or used by each task.
- Relevant work from earlier this year with a similar objective of measuring multimodal interactions with no textual shortcuts: `BEARCUBS: A benchmark for computer-using web agents`

---

> ### Author Response · Authors · 2025-11-26
> **Official Comment by Authors (1/3)**
>
> > W1: There is missing a human baseline to calibrate what model accuracy means. It would provide an estimate for the performance ceiling of this task.
>
> A: Thank you for this constructive suggestion. We have conducted an additional human evaluation to establish a baseline and calibrate model performance. Specifically, we recruited 6 human annotators, all of whom are Master's students specializing in AI. We distributed the full MM-BrowseComp dataset evenly among them. Each question was subject to a 10-minute time limit, during which annotators were allowed to use standard web browsing tools but were prohibited from using AI assistants. The resulting human performance is shown below:
>
> | Metric               | OA(%)  | SA(%) | AVG CS(%) | Avg. Cost Time (mins) |
> |----------------------|--------|-------|--------|---------------------|
> | Human Performance    | 60.26 | 50.00   | 58.36 | 7.8                 |
>
>
> This evaluation indicates that **MM-BrowseComp tasks present challenges even for humans**. Furthermore, it demonstrates that there remains a **significant margin for improvement before agents can match human-level capabilities**.
>
> ---
>
>
> > W2: In 3.1.1, authors assert that essential information to solve a task should not appear in any text source. However, there is no mention of how this verification is done.
>
> A:  Thank you for your insightful question. We ensure that no textual shortcuts exist by verifying that at least one checklist item per question is grounded strictly in non-textual modalities content. This is achieved through a two-step process.
>
> First, during annotation, we instruct human experts to intentionally ensure that at least one essential information (checklist item) required to solve the task is derived from multimodal content. Additionally, they are required to **use various tools, such as search engines, to verify that this information is not present in any text-based** sources available on the web.
>
> Second, during validation, a core team performs a rigorous check using tools like OpenAI DeepResearch. We specifically investigate if the reference links provided by models contain "leaked" answers in text form. Through this multi-round validation, **25.3% of the original 300 questions were discarded and 21.0% were significantly modified**, leaving only 224 instances where we are confident that the solution relies on unavoidable multimodal reasoning.
>
> ---
>
>  > W3: Although the authors repeatedly refer to “video-dependent” tasks, the paper never specifies how models are expected to engage with videos. Are agents intended to interact with video content directly, or are they simply expected to rely on accompanying textual information, like transcripts or descriptions?
>
> A: Thank you for this insightful question. Regarding video-dependent tasks, they are treated the same way as image-dependent tasks. We only provide questions and do not offer any additional information during the problem-solving process. Therefore, how an agent interacts with video content is entirely up to the agent's architecture and toolset.
>
> We have observed that **different agents exhibit unique behaviors and use custom tools to achieve their goals**. For example, Gemini models access to a Google in-house YouTube tool, which allows them to retrieve video frames from YouTube videos. Similarly, OpenAI o3 and Agent-R1 uses a code executor and file system to download images and videos from the web into a file sandbox, enabling it to read and understand multimodal content. As such, how an agent engages with video content depends entirely on its specific design. What we ensure is that there are no textual shortcuts for each question.

---

> ### Author Response · Authors · 2025-11-26
> **Official Comment by Authors (2/3)**
>
> > Q1: Figure 3 could be clearer. Rather than focusing on whether each input includes an image, it might be more informative to present statistics on the actual modalities required or used by each task.
>
> A: Thank you very much for your constructive suggestion. We have conducted a detailed analysis of the actual modalities required for each subtask in the table below. And this table will be included in **Appendix B** due to space limits.
>
> | Subtask                            | Samples | Total_Checklist_items | Text_item | Image_item | Video_item |
> |------------------------------------|---------|-----------------------|------------|-------------|-------------|
> | Academic Chart Q&A                 | 10      | 35                    | 25         | 10          | 0           |
> | Academic Paper Q&A                 | 12      | 45                    | 22         | 22          | 1           |
> | Biological Image Reasoning         | 10      | 42                    | 15         | 13          | 14          |
> | Cultural Reasoning                 | 7       | 25                    | 9          | 11          | 5           |
> | Electronics Detail Q&A             | 11      | 13                    | 4          | 4           | 5           |
> | Electronics Identification         | 11      | 22                    | 0          | 12          | 10          |
> | Equipment Fault Diagnosis          | 9       | 23                    | 11         | 2           | 10          |
> | Film & TV Reasoning                | 12      | 37                    | 11         | 2           | 24          |
> | Financial Analysis                 | 11      | 37                    | 21         | 14          | 2           |
> | Financial Chart Analysis           | 10      | 31                    | 7          | 17          | 7           |
> | Geographic Map Reasoning           | 8       | 24                    | 2          | 11          | 11          |
> | Historical Archive Reasoning       | 12      | 38                    | 13         | 21          | 4           |
> | Time-sensitive Media Reasoning     | 10       | 28                    | 8          | 10          | 10          |
> | Time-sensitive Media Recognition   | 9      | 27                    | 12         | 1           | 14          |
> | Tutorial Video Reasoning           | 10      | 31                    | 12         | 0           | 19          |
> | Vehicle Detail Q&A                 | 9       | 18                    | 9          | 5           | 4           |
> | Vehicle Identification             | 11      | 24                    | 2          | 13          | 9           |
> | Video Game Walkthroughs            | 12      | 31                    | 19         | 3           | 9           |
> | Visual Fault Diagnosis             | 8       | 19                    | 1          | 11          | 7           |
> | Visual Game Q&A                    | 12      | 40                    | 13         | 14          | 13          |
> | Visual Geolocalization & Reasoning | 11      | 38                    | 16         | 14          | 8           |
> | Visual Navigation                  | 9       | 30                    | 7          | 20          | 3           |
> | TOTAL                              | 224     | 658                   | 239        | 230         | 189         |
>
>
> The results demonstrate that we have achieved a highly balanced distribution across the three modalities within the total checklist items, comprising **239 Text**, **230 Image**, and **189 Video** items. Furthermore, nearly every subtask incorporates items from all three modalities. The few exceptions arise only where the **intrinsic nature of the subtask definition** precludes specific modalities (e.g., the 'Tutorial Video Reasoning' subtask naturally excludes static image inputs).

---

> ### Author Response · Authors · 2025-11-26
> **Official Comment by Authors (3/3)**
>
> > Q2: Relevant work from earlier this year with a similar objective of measuring multimodal interactions with no textual shortcuts: BEARCUBS: A benchmark for computer-using web agents
>
> A: Thank you for pointing out this important work! **We have cited this reference in the updated PDF**. While both benchmarks involve multimodal web agents, they **differ significantly in motivation, target agents, data diversity, and evaluation metric:**
>
> - **Motivation & Target Agents**: BEARCUBS focuses on evaluating the ability of agents to interact with live web GUIs. It is best suited for **"Computer Use" or "Operation-centric" agents**. In contrast, MM-BrowseComp targets **"Deep Research" agents** capable of synthesizing information from hundreds of web pages using various tool APIs and parallel queries. Our focus is on reasoning and multimodal retrieval rather than UI navigation.
>
> - **Data Diversity**: MM-BrowseComp offers broader coverage. While BEARCUBS contains 111 questions (roughly half text-based and half multimodal), MM-BrowseComp comprises **224 questions** spanning five distinct domains: Media, Technology, Society, Geography, and Academics. Our dataset places a stronger emphasis on integrating evidence from text, images, and videos across the open web to perform complex multi-hop reasoning.
>
> - **Evaluation Metrics**: BEARCUBS relies primarily on the correctness of the final answer. Conversely, MM-BrowseComp introduces a **fine-grained checklist evaluation**. This approach differentiates between merely guessing the correct answer and rigorously completing the retrieval and reasoning process, offering a more precise measure of an agent's multimodal capabilities.
>
> Thank you for pointing out this important work again. We have cited this reference in the updated PDF.

---

### Official Review · Reviewer_cbDe · 2025-11-01

**Soundness:** 3
**Presentation:** 3
**Contribution:** 2
**Rating:** 6
**Confidence:** 3

**Summary:**

This paper introduces MM-BrowseComp, a new benchmark designed to evaluate multimodal browsing agents that integrate reasoning and tool use. It comprises 224 hand-crafted questions across 22 subtasks, requiring retrieval and reasoning over both textual and visual information. Each question includes a verified checklist that tracks reasoning steps, allowing fine-grained analysis beyond final-answer accuracy. Experimental results show that even state-of-the-art models like OpenAI’s o3 achieve an accuracy of 29%, highlighting the difficulty of multimodal reasoning and the limitations of current models. Overall, the paper provides a challenging dataset that fills an important gap in multimodal browsering agent evaluation.

**Strengths:**

* The proposed benchmark is constructed through multiple rigorous verification phases.
* The experiment part systematically compares a wide range of state-of-the-art closed- and open-source models, offering a clear view of current limitations and performance gaps.

**Weaknesses:**

* The tasks in this benchmark are often intentionally complex and involve multi-hop reasoning, which may not accurately reflect the typical multimodal search behaviors encountered in real-world web browsing scenarios.
* The heavily hand-crafted nature of the benchmark may limit real-world generalizability.

**Questions:**

See weakness.

---

> ### Author Response · Authors · 2025-11-26
> **Official Comment by Authors**
>
> We sincerely appreciate your positive assessment of our work. We are encouraged by your recognition that MM-BrowseComp provides a clear view of the limitations on current browsing agents. We will address your concerns in detail below.
>
> > W1: The tasks in this benchmark are often intentionally complex and involve multi-hop reasoning, which may not accurately reflect the typical multimodal search behaviors encountered in real-world web browsing scenarios.
>
> A: Thank you for this constructive feedback. We acknowledge that the tasks in MM-BrowseComp are indeed more complex than daily multimodal search scenarios. However, this is a deliberate design choice essential for the following reasons:
>
> 1. **Distinguishability**: With the advent of powerful reasoning models such as OpenAI o3, DeepSeek-R1, and Gemini-2.5 Pro, current agent benchmarks (e.g., GAIA, MMSearch) are becoming saturated, as shown in Figure 1. Therefore, the community urgently needs a more challenging benchmark to distinguish the capabilities of frontier agents and determine their performance upper bounds.
>
> 2. **Generalizability**: Achieving high performance on complex tasks of MM-BrowseComp implies robust core capabilities in reasoning, planning, and multimodal retrieval and comprehension. We believe these core skills generalize well to simpler, real-world tasks, analogous to how mastering AIME[1] problems implies proficiency in standard K-12 mathematics. Similar design philosophies are seen in benchmarks like BrowseComp [2] and HLE [3]: while they do not mirror typical daily routines, they are widely recognized as valuable for characterizing the limits of model capabilities.
>
> 3. **Beyond Routine Queries**: Fundamentally, we argue that agents should not be limited to mimicking human daily queries. The ability to solve complex tasks paves the way for more powerful autonomous copilot systems, enabling them to handle future challenges that extend far beyond simple information retrieval.
> ---
>
> > W2: The heavily hand-crafted nature of the benchmark may limit real-world generalizability.
>
> A: Thank you for your valuable feedback. As discussed in our response to W1, the intentional difficulty of our hand-crafted questions allows for better **distinguishability** and pushes models **beyond routine queries**. While typical real-world web searches may be less complex, high performance on our hand-crafted challenging tasks demonstrates a higher upper bound in core capabilities, such as reasoning, planning, and multimodal retrieval. We argue that an agent possessing these robust core skills is inherently capable of **generalizing to (and easily solving) simpler real-world general queries**.
>
> Additionally, although MM-BrowseComp is hand-crafted, it is designed to be **highly diverse and representative**, spanning 5 categories and 22 subtasks. Furthermore, each task was **constructed by experienced domain experts** to ensure quality and relevance. This combination of diverse data sources and high-quality expert curation ensures that MM-BrowseComp remains robust and generalizes well to real-world scenarios.
>
> Below, we list several benchmarks published in top-tier conferences which are hand-crafted and challenging:
>
> | Benchmark          | Venue         | # Instances       | # Categories | Reference |
> |--------------------|---------------|-------------------|-------------|-----------|
> | GAIA               | ICLR 2024     | 165 (val set)     | 3           | https://iclr.cc/virtual/2024/poster/18176 |
> | MMSearch           | ICLR 2025     | 300               | 14          | https://iclr.cc/virtual/2025/poster/30134 |
> | AgentHarm          | ICLR 2025     | 440               | 11          | https://iclr.cc/virtual/2025/poster/32106 |
> | OSWorld            | NeurIPS 2024  | 369               | 4           | https://neurips.cc/virtual/2024/poster/97468 |
> | Mind2Web 2         | NeurIPS 2025  | 130               | -           | https://neurips.cc/virtual/2025/loc/san-diego/poster/121798 |
> | Open CaptchaWorld  | NeurIPS 2025  | 225               | 20          | https://neurips.cc/virtual/2025/poster/121537 |
> | BEARCUBS           | COLM 2025     | 111               | 1           | https://openreview.net/forum?id=0JzWiigkUy |
>
>
> [1] American invitational mathematics examination - AIME. MAA. 2025
>
> [2] Browsecomp: A simple yet challenging benchmark for browsing agents, OpenAI, arxiv, 2025.
>
> [3] Humanity's last exam. Scale AI. arXiv. 2025.

---

### Note · Authors · 2026-01-06

I have read and agree with the venue's withdrawal policy on behalf of myself and my co-authors.